# Stochastic gene expression in proliferating cells: Differing noise intensity in single-cell and population perspectives

**Zhanhao Zhang**[1☺], **Iryna Zabaikina**[2,3☺], **Cesar Nieto** (iD)[1☺], **Zahra Vahdat**[1,4,5], **Pavol Bokes**[2], **Abhyudai Singh** (iD)[6*]

**1** Department of Electrical and Computer Engineering, University of Delaware, Newark, Delaware, United States of America, **2** Department of Applied Mathematics and Statistics, Comenius University, Bratislava, Slovakia, **3** Department of Mathematical Analysis and Numerical Mathematics, Comenius University, Bratislava, Slovakia, **4** Dan L. Comprehensive Cancer Center, Baylor College of Medicine, Houston, Texas, United States of America, **5** Department of Molecular and Cellular Biology, Baylor College of Medicine, Houston, Texas, United States of America, **6** Department of Electrical and Computer Engineering, Biomedical Engineering, Mathematical Sciences, Center of Bioinformatics and Computational Biology, University of Delaware, Newark, Delaware, United States of America

☺ These authors contributed equally to this work.
* absingh@udel.edu

**Data availability statement:** The simulation code used for these plots is publicly available at https://doi.org/10.5281/zenodo.10884051.

## Abstract

Random fluctuations (noise) in gene expression can be studied from two complementary perspectives: following expression in a single cell over time or comparing expression between cells in a proliferating population at a given time. Here, we systematically investigated scenarios where both perspectives can lead to different levels of noise in a given gene product. We first consider a stable protein, whose concentration is diluted by cellular growth. This protein inhibits growth at high concentrations, establishing a positive feedback loop. Using a stochastic model with molecular bursting of gene products, we analytically predict and contrast the steady-state distributions of protein concentration in both frameworks. Although positive feedback amplifies the noise in expression, this amplification is much higher in the population framework compared to following a single cell over time. We also study other processes that lead to different noise levels even in the absence of such dilution-based feedback. When considering randomness in the partitioning of molecules between daughters during mitosis, we find that in the single-cell perspective, the noise in protein concentration is independent of noise in the cell cycle duration. In contrast, partitioning noise is amplified in the population perspective by increasing randomness in cell-cycle time. Overall, our results show that the single-cell framework that does not account for proliferating cells can, in some cases, underestimate the noise in gene product levels. These results have important implications for studying the intercellular variation of different stress-related expression programs across cell types that are known to inhibit cellular growth.

**Funding:** P.B. and I.Z. have been supported by the Slovak Research and Development Agency under contract No. APVV-23-0039 and the VEGA grant 1/0755/22. A.S. and C.N. have been supported by NIH-NIGMS through grant R35GM148351. The funders had no role in study design, data collection and analysis, decision to publish, or preparation of the manuscript.

**Competing interests:** The authors have declared that no competing interests exist.

## Author summary

Expression levels of gene products can exhibit cell-to-cell variation within an otherwise genetically identical cell population. Such random variation, often referred to as expression noise, has been reported in all organisms, from bacterial to human cells. This contribution quantifies the degree of random fluctuations in the concentration of a specific protein from two complementary perspectives: following its concentration in a single cell over time (single-cell perspective) and across a cell population (population perspective). When are the statistical fluctuations in expression levels different between these two perspectives? Analytical results combined with agent-based models that track protein concentration in each cell of a growing colony identify two such scenarios. The first scenario corresponds to high levels of a specific protein reducing cellular growth. In the second scenario, the random segregation of molecules between daughters (partitioning) dominates expression noise. In both these cases, the classical single-cell approach underestimates the degree of concentration fluctuations seen across a cell population. These results have important implications for regulating stochastic variations in diverse stress-related expression programs that promote drug-tolerant states in microbial and cancer cells.

## Introduction

The intracellular level of gene products is the result of complex interconnected biochemical processes that are intrinsically stochastic and often operate with low-copy number components. This stochasticity is manifested as intercellular variation in gene expression levels within an isogenic cell population despite controlling for factors, such as the extracellular environment and cell-cycle effects [1–4]. Random fluctuations (noise) in gene expression levels fundamentally impact all aspects of cell physiology and the fidelity of cellular information processing. Not surprisingly, depending on the gene function and context, expression noise is subject to evolutionary pressures [5–10] and actively regulated through diverse mechanisms. For example, the promoter architecture/genomic environment [11–13], the kinetics of different gene expression steps [14–16], the inclusion of feedback/feedforward loops [17–23], and the ubiquitous binding of proteins to *decoy sites* [24–26] have been shown to both attenuate or amplify noise levels.

Over the last few decades, single-cell studies have revealed beneficial roles of noise in gene product levels. These include, but are not limited to, driving genetically identical cells to different fates [27–34] and facilitating population adaptation to environmental fluctuations [35–39]. The latter scenario is exemplified by rare populations of clonal cells that survive lethal stresses, as seen in antibiotic treatment of bacteria [40–45], or cancer cells undergoing chemotherapy [46–50]. The non-genetic basis of heterogeneous single-cell responses to stress is a topic of current research, and several publications have implicated preexisting drug-tolerant expression states arising as result of noise in gene regulatory networks [51,52].

Random fluctuations in the level of a given protein can be studied from two perspectives: single cell and population [53–55]. The single-cell perspective approach captures the stochastic dynamics of protein level in a single cell over time (Fig 1A), and here the effects of cell growth and division are either ignored or implicitly captured (for example, through continuous dilution of concentration). This framework is also known as *single lineage* since it randomly tracks one of the branches of a lineage tree [53]. In the population perspective, one

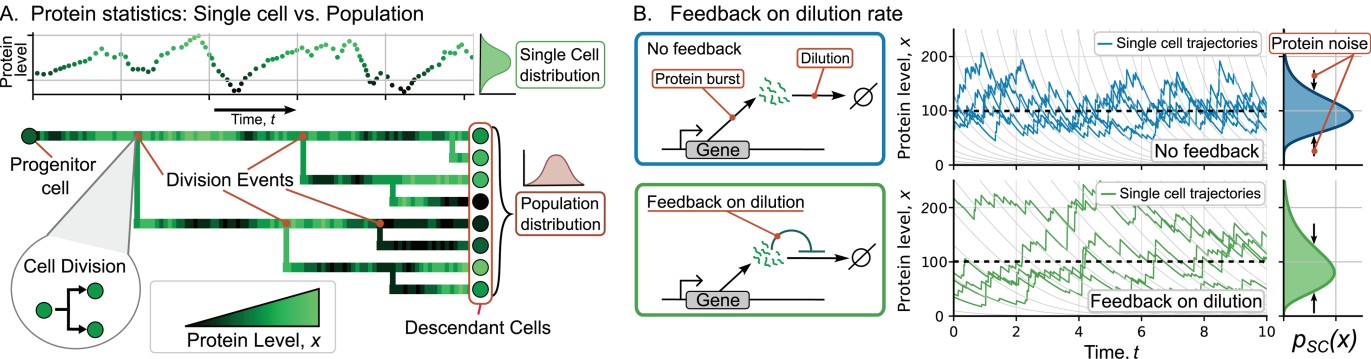

**Fig 1. Single-cell and population perspectives for investigating stochastic gene expression with dilution-based feedback regulation.** (**A**) *(top:)* For the single-cell perspective, concentration of a given protein is tracked along a single lineage. *(bottom:)* From a population point of view, the protein concentration distribution is obtained across all descendants of the colony. Different shades of green represent the protein level for each cell. (**B**) *(left:)* Schematic of the gene expression model with random bursts of protein synthesis, and concentration dilution in between burst events. In the model without regulation (blue), the dilution rate is constant. In the model with feedback on dilution (green), the dilution rate decreases as the protein concentration increases, according to (3). *(right:)* Sample trajectories of protein concentration in a single cell, along with the corresponding single-cell protein concentration distributions for both models. The gray lines in the background show protein dilution trajectories; horizontal dashed lines represent the mean concentration in both models. These trajectories are for different single cells with initial values selected from previous simulations such that they start from steady state conditions. Parameter values used for these trajectories are $\beta = 10$, $k = 1/100$, $\lambda$ is found using (7) with the mean protein level set to 100. Therefore, $\lambda = 10$ (no feedback), $\lambda = 4.76$ (with feedback). Time is presented in units such as $\gamma = 1$.

explicitly considers an exponentially expanding cell population, and gene product variability is quantified across all cells at a given time point. A fundamental question of interest is *when do these complementary perspectives predict different degrees of stochastic variation in gene expression?*

Our analysis identifies two scenarios where single-cell and population perspectives yield different extents of fluctuations in the concentration of a given protein of interest. *The first scenario arises when the intracellular concentration directly or indirectly affects cellular growth*, and hence determines the cell's proliferation capacity. We specifically focus on the case where high protein concentration inhibits cellular growth. This drives the concentration even higher because of reduced dilution. This effect implements a positive feedback loop [56–60]. This expression-growth coupling can be seen in many cases of protein-induced stress response [61], cell resource saturation [62,63], and is a feature of many stress-tolerant expression programs. For example, high expression of specific proteins comes at the cost of inhibiting cellular growth in the absence of stress, but improves cell survival in the presence of stress [58,64–66].

From a mathematical perspective, for this scenario we propose approaches based on the solution of the associated differential Chapman-Kolmogorov equation (dCKE), and the population balance equation (PBE), to derive protein concentration distributions in the single-cell and population perspectives, respectively. The simplicity of our modeling frameworks allows exact derivations of the corresponding probability density function (pdf), which are then compared and contrasted between the two perspectives with increasing feedback strength.

*The second scenario corresponds to randomness in the partitioning of protein molecules between two daughters during mitosis and cytokinesis* [67]. In this scenario, concentration fluctuations in the single-cell perspective are modeled using the formalism of Stochastic Hybrid Systems (SHS) resulting in an exact analytical formula for the concentration noise level, as quantified by the steady-state squared coefficient of variation of protein concentration. The corresponding statistics from the population perspective are obtained via agent-based models that track expression levels within each cell of a proliferating colony. Interestingly, our results

show that coupling the partitioning process with the inherent randomness of cell-cycle times enhances expression variability across the population as compared to the single-cell perspective. We begin by formally introducing the two different perspectives for studying stochastic expression, and how stochasticity is modeled based on random bursts of gene activity.

## Results

### Single-cell and population level analysis to quantify gene expression statistics

We study stochastic variations in the concentration of a specific protein within proliferating cells using two complementary perspectives. These perspectives are graphically illustrated in Fig 1A, where the expansion of a cell colony is represented as a lineage tree: the root of the tree represents the progenitor cell, each branching point is a cell division event, and the horizontal distance (time) between consecutive branching points represents the cell-cycle duration. The color intensity represents the protein level at a given time: the lighter the green, the higher the protein level. The color intensity in Fig 1A bottom can be related to a two-dimensional graph of protein level over time (Fig 1A top). During cell division, a mother cell splits into two identical daughters, each inheriting half of the mother's volume and protein amount. Thus, the protein concentration in the newborn daughters is assumed to be equal to the mother's concentration just before division. This assumption of perfect partitioning will be relaxed later in the manuscript. In the single-cell approach, only one of the two daughter cells is tracked after division and protein statistics are determined based on multiple single-lineage paths over time. In contrast, in the population approach, both daughter cells are tracked, and statistics are estimated on all descendant cells at a fixed point in time.

**Ergodicity of protein distributions.** In our approach, we consider two main methods for estimating protein statistics (mean and noise). The first method involves taking the time-averaged statistics of the stochastic process over a long trajectory (as shown in Fig 1A top in which the statistics are obtained from a single trajectory). The second method involves estimating the statistics at a given time across different replicas (as illustrated in Fig 1B in which the statistics are estimated from multiple independent trajectories at the last point). From a mathematical perspective, when these two averaged values are equivalent, the stochastic process is considered ergodic [68]. Ergodicity will be used for our analytical derivations by assuming that our statistics are time-independent. Due to the computational expense of following a proliferating population for a prolonged period, our simulations' results, unless otherwise specified, will be estimated using the second method. For single cells, we obtain statistics from thousands of independent cells at a given time. For the population perspective, we perform the statistics on all cells across thousands of colonies at a given time.

**Gene expression as a burst-dilution process.** To analytically derive and contrast the concentration distribution in both perspectives, we take advantage of a simple model of gene expression that has previously been introduced [69] and validated with single-cell data [70, 71]. This model consists of modeling protein synthesis as occurring in short periods of intense gene activity, often referred to in the literature as *bursting*. Diverse mechanisms that span all stages of gene expression (promoter activation/transcription/translation) have been simplified as effective bursting processes [72–79]. During each burst event, protein concentration increases by a random amount. Considering a long-lived protein (i.e., half-life much longer than the cell-doubling time), the effects of cell growth are captured through continuous dilution along the cell cycle. Given that prokaryotes typically have short cell doubling times, our model is primarily applicable to the long-lived proteins found in these microorganisms [80].

### Effects of growth-mediated feedback on protein concentration fluctuations

In this section, we describe models that couple bursting expression events with dilution-based positive feedback, considering both single-cell and population perspectives. We analyze these models to derive *exact* steady-state distributions for protein concentration. These distributions are then compared and contrasted between the two perspectives, with and without feedback regulation.

**Model description.**   In the single-cell perspective, the protein concentration $x(t)$ at time $t$ within an individual cell evolves stochastically according to the following rules. Burst events occur according to a Poisson process with rate or burst frequency $\lambda$. During each *burst*, $x$ increases instantly with a burst size $b$ drawn from an exponential distribution with mean $\beta$. These increments in concentration are conveniently represented by the reset:

$$x \xrightarrow{\lambda} x + b, \quad b \sim Exp(1/\beta). \tag{1}$$

It is important to point out that both the burst frequency and size in this concentration model are invariant with respect to the cell size. This implicitly assumes appropriate scaling of expression rates (in terms of the number of molecules synthesised per unit time) with cell size [81–89].

Feedback in dilution is modeled phenomenologically by considering the cellular growth rate

$$\frac{\gamma}{1 + kx}, \tag{2}$$

which is a decreasing function of concentration $x$ (Fig 1B, lower panels). This results in the following dilution dynamics in between burst events

$$\frac{\mathrm{d}x}{\mathrm{d}t} = -\frac{\gamma x}{1 + kx}, \tag{3}$$

where $k \geq 0$ can be interpreted as the *feedback strength* and $\gamma > 0$ is the *maximum dilution rate*. The growth rate also defines the rate with which cell division occurs. In our model, division is defined as a point process with a propensity given by (2) with the sole effect, at the population level, of generating a new cell with the same protein level of the parent cell. Therefore, in the single-cell approach, division has no effect on protein concentration. In more mathematical terms, intracellular fluctuations in protein concentration are captured by the piecewise deterministic Markov process (PDMP) $x(t)$ defined by (1)–(3). For the reader's convenience, we provide a compilation of model parameters and symbols used to quantify concentration statistics in Table 1.

**Protein statistics with no feedback.**   Before analyzing the feedback model, we briefly review the special case of no feedback ($k = 0$) that corresponds to constant cellular growth and dilution with rate $\gamma$ (Fig 1B, upper panels). Prior analysis of this unregulated gene expression model predicted that the steady-state protein distribution $p(x)$ followed a gamma distribution with parameters $\lambda/\gamma$ and $\beta$ (shape and scale, respectively) [69], consistent with single-cell variations in the expression of specific proteins in *Escherichia coli* and *Saccharomyces cerevisiae* [71]. The steady-state protein distribution can be analyzed through its statistical properties (defined in Table 1): the mean $\overline{\langle x \rangle}$, the squared coefficient of variation $CV_x^2$ (quantifying the noise in protein level), and the skewness (measure of the distribution asymmetry). For

**Table 1. Model parameters and variables studied in the text.**

| Notation | Interpretation |
|---|---|
| $x$ | Random process representing the intracellular protein concentration |
| $\lambda$ | Rate of occurrence of protein bursts, i.e., burst frequency |
| $\beta$ | Mean size of exponentially distributed protein bursts |
| $\gamma$ | Maximum dilution rate |
| $k$ | Feedback strength quantifying the expression-dilution coupling |
| $\varepsilon$ | Degree of partitioning noise in the segregation of protein molecules between daughters |
| $\tau_d$ | Cell cycle duration |
| $x^+$ | Protein concentration just after a cell division. |
| $\langle x^n \rangle$ | $n$-th order moment of a random variable/process, $\langle \cdot \rangle$ is the expectation operator |
| $\overline{\langle x^n \rangle} := \lim_{t\to\infty} \langle x^n \rangle$ | Steady-state moment of a random variable |
| $\overline{\langle x \rangle}$ | Steady-state mean protein concentration |
| $\sigma_x^2 := \overline{\langle x^2 \rangle} - \overline{\langle x \rangle}^2$ | Steady-state variance of protein concentration |
| $\mathrm{CV}_x^2 := \sigma_x^2 / \overline{\langle x \rangle}^2$ | Steady-state coefficient of variation of protein concentration, also defined as noise |
| $\mathrm{Skew}_x := \left( \overline{\langle x^3 \rangle} - 3\overline{\langle x \rangle}\sigma_x^2 - \overline{\langle x \rangle}^3 \right)/\sigma_x^3$ | Steady-state skewness of protein concentration |
| $(\cdot)_{SC}, \ (\cdot)_{Pop}$ | Statistics of protein concentration in single cell and population perspectives, respectively |

this unregulated case ($k = 0$), we obtain

$$\overline{\langle x \rangle} = \frac{\lambda\beta}{\gamma}, \quad \mathrm{CV}_x^2 = \frac{\beta}{\langle x \rangle}, \quad \mathrm{Skew}_x = 2\sqrt{\frac{\beta}{\langle x \rangle}}, \tag{4}$$

and it is interesting to note the ratio $\mathrm{Skew}_x/\mathrm{CV}_x = 2$.

   **Protein distribution from a single-cell perspective.**  For the model with feedback ($k>0$), the time evolution of the probability density function (pdf) $p_{SC}(x,t)$ is given by the differential Chapman-Kolmogorov equation (dCKE) [69,90,91] (see Methods). Here and throughout the article the subscript $SC$ is used to denote distributions and statistics in the single-cell perspective. The stationary protein distribution $p_{SC}(x)$ defined as $p_{SC}(x) := \lim_{t\to\infty} p_{SC}(x,t)$ satisfies

$$p_{SC}(x) = (1 + kx)\frac{\beta\eta^2}{\Gamma(\lambda/\gamma)} e^{-\eta x}(\eta x)^{\lambda/\gamma - 1}, \quad \eta = 1/\beta - \lambda k/\gamma, \tag{5}$$

where $\Gamma(z) = \int_0^\infty u^{z-1} e^{-u} \mathrm{d}u$ is the gamma function (see S1 Text Sect 1 for details on dCKe and its analytical solution) and $z = \lambda/\gamma$ is its argument. We observe from (5) that $p_{SC}(x)$ exists only for the set of parameters $\lambda$, $\beta$, $\gamma$, and $k$ that satisfy $\eta > 0$. An interpretation of this condition is that the average production rate $\lambda\beta$ must be less than the maximum dilution rate $\gamma/k$ in (3). Otherwise, the protein dilution is not fast enough to compensate for the protein production

rate and the mean level unboundedly increases over time; thus, the stationary distribution does not exist. Fig 2A shows the space of parameters where the distribution (5) exists.

**Moments of protein distribution at single-cell perspective.** Using (5), we obtain the following statistical properties of the protein level in terms of the feedback strength $k$:

$$\overline{\langle x \rangle}_{SC} = \overline{\langle x \rangle} \frac{1 + k\beta}{1 - k\overline{\langle x \rangle}}, \tag{6a}$$

$$\left(CV_x^2\right)_{SC} = \frac{\beta}{\overline{\langle x \rangle}}\left(1 - \frac{1 + \lambda/\gamma}{\left(1 + \frac{1}{k\beta}\right)^2}\right), \tag{6b}$$

$$(Skew_x)_{SC} = 2\frac{(\lambda/\gamma + 1 - (\eta\beta)^3)}{(\lambda/\gamma + 1 - (\eta\beta)^2)^{3/2}}, \tag{6c}$$

where $\overline{\langle x \rangle}$ is the mean concentration of the unregulated process as per (4). To have a fair comparison of the effect of increasing $k$, we hold the mean protein level $\overline{\langle x \rangle}_{SC}$ fixed for different values of $k$. To achieve this, we use (6a) to express the burst frequency $\lambda$ as a function of $k$:

$$\lambda = \frac{\gamma\overline{\langle x \rangle}_{SC}}{\beta(1 + k\beta + k\overline{\langle x \rangle}_{SC})}. \tag{7}$$

As a graphic example, the way $\lambda$ must change as we increase $k$ to maintain $\overline{\langle x \rangle}_{SC} = 100$ is shown in Fig 2A (red line). Fig 2B (*top*) shows that under weak feedback, there are minimal differences between single-cell and population perspectives. However, a significant divergence emerges under strong feedback, although the mean concentration remains fixed

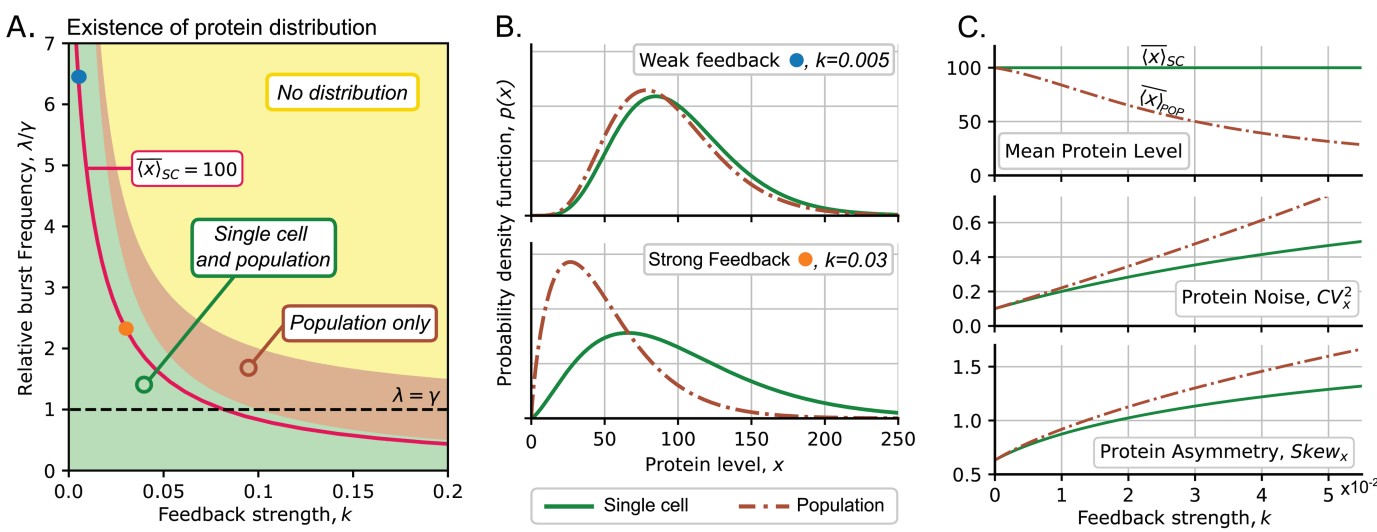

**Fig 2. Comparison of protein distribution in single-cell and population perspectives as the feedback intensifies.** (**A**) The region of existence of the steady-state protein distribution in terms of the feedback strength $k$ and relative burst frequency $\lambda/\gamma$. *Green region*: stationary distribution $p(x)$ exists in both single cell and population perspectives. *Brown region*: Only the distribution in the population perspective exists. *Yellow region*: Distribution does not exist in any of the frameworks. *Bold red line*: a set of values $(k, \lambda/\gamma)$, resulting in fixed $\overline{\langle x \rangle}_{SC} = 100$ as per (7). Black dashed line represents the maximum burst frequency in which the distribution exists for both perspectives regardless the feedback strength. (**B**) Comparison of protein distribution in single-cell (solid green line) and population perspectives (brown dashed) as feedback increases: *(top:)* weak feedback, *(bottom:)* strong feedback. (**C**) *From top to bottom:* Mean protein level, protein noise, and distribution asymmetry in the single cell and population frameworks; $\lambda$ is chosen so that $\overline{\langle x \rangle}_{SC} = 100$ as $k$ increases following the bold red line in panel (**A**). For all plots, we set $\beta = 10$, $\gamma = 1$.

as per (7) (Fig 2B *bottom*). It is also possible to notice that for a giving $\overline{\langle x \rangle}_{SC}$, $\left(CV_x^2\right)_{SC}$, and $(\text{Skew}_x)_{SC}$ are increasing functions of $k$ (Fig 2C green lines). Thus, as feedback becomes stronger, the protein distribution exhibits greater noise and becomes more right-skewed. In particular, in the limit of weak feedback strength ($k \ll 1$), the distribution statistics are approximated as:

$$\left(CV_x^2\right)_{SC} \approx \frac{\beta}{\langle x \rangle_{SC}} \left(1 + k\left(\beta + \overline{\langle x \rangle}_{SC}\right)\right), \tag{8a}$$

$$(\text{Skew}_x)_{SC} \approx 2\sqrt{\frac{\beta}{\langle x \rangle_{SC}}} \left(1 + \frac{k}{2}\left(\beta + \overline{\langle x \rangle}_{SC}\right)\right). \tag{8b}$$

Notice that without regulation ($k = 0$), the protein distribution in a single cell becomes identical to the unregulated one with statistics given by (4). The presence of weak regulation perturbs these unregulated statistics (4) by an increasing function of $k$. Finally, recall that the ratio $(\text{Skew}_x)_{SC} / (CV_x)_{SC}$ was equal to 2 for unregulated gene expression ($k = 0$), but decreases below 2 with increasing positive feedback strength $k$.

**Protein distribution for population perspective.** To extend the single-cell framework to a population one, it is required to describe the dynamics of cell proliferation. We assume that cell division events are modeled by a non-homogeneous Poisson process with rate $\gamma/(1 + kx)$ [92]. Then, a cell with protein level $x$ at time $t$ has a probability $[\gamma/(1 + kx)]dt$ to divide during the next infinitesimal time interval $(t, t + dt)$. It also follows that cells with low protein concentrations proliferate faster than those with high protein levels. Note that in the limit of the unregulated protein ($k = 0$), division events occur according to the standard Poisson process with rate $\gamma$, and the cell cycle is exponentially distributed with mean $1/\gamma$.

In the population framework, in addition to the protein, we also quantify the time evolution of the cell population size. To this end, we introduce the population density function $h(x,t)$, which describes the population as the number of cells with a given concentration $x$ at time $t$. To obtain $h(x,t)$ we solve the associated population balance equation (PBE) [55,93] (see Methods). The PBE is similar to the dCKE, but after division, the process follows the dynamics of both daughter cells. In steady-state conditions, only the population size grows, whereas the proportion of cells with a given protein level remains steady. Thus, $h(x,t)$ can be decomposed as

$$h(x,t) = f(t)p_{Pop}(x), \tag{9}$$

where $f(t)$ is an exponential function (explained in S1 Text Sect 2) associated with the population size growth and $p_{Pop}(x)$ is the stationary probability density function of the protein concentration $x$; the subscript *Pop* indicates quantities determined in the population perspective. In S1 Text Sect 2, we also show that $p_{Pop}(x)$ has the closed expression:

$$p_{Pop}(x) = (1 + kx)\frac{\beta\rho^2}{\Gamma(\xi)}e^{-\rho x}(\rho x)^{\xi-1}, \quad \rho = \frac{1}{\beta} - k\xi, \quad \xi = \frac{\lambda/\gamma}{k\beta + 1}. \tag{10}$$

Similarly to the single-cell approach, the protein distribution at population perspective exists if $\rho > 0$. Fig 2A shows that the population distribution exists whenever the single-cell distribution does. Moreover, $p_{Pop}(x)$ always exists when $\lambda < \gamma$, that is, when the burst frequency is below the maximum dilution rate. In general, the population distribution exists for values of $\lambda$ that satisfy $(\lambda - \gamma)\beta < \gamma/k$. This existence condition is less strict than the one for a

single-cell model: a population distribution may exist even if the single-cell distribution does not (Fig 2A).

**Moments of protein concentration for population perspective.** A comparison of the protein distributions for both frameworks is shown in Fig 2B. If the feedback is weak (upper panel), then the difference between the distributions is insignificant; this is consistent with the fact that both distributions are identical to the gamma distribution in the unregulated case ($k = 0$). As the feedback becomes stronger (lower panel), the differences between protein distributions become larger: population distribution shifts to lower concentration and becomes more light-tailed, indicating a larger portion of cells with low protein concentration.

The statistics of the protein level in the population perspective are obtained as:

$$\overline{\langle x \rangle}_{Pop} = \overline{\langle x \rangle} \frac{1 + k\beta}{1 + k\beta - k\overline{\langle x \rangle}}, \tag{11a}$$

$$\left(\mathrm{CV}_x^2\right)_{Pop} = \frac{\beta}{\overline{\langle x \rangle}} \left(1 + \tilde{k} - \frac{\lambda}{\gamma}\tilde{k}^2\right), \tag{11b}$$

$$(\mathrm{Skew}_x)_{Pop} = 2\sqrt{\frac{\beta}{\overline{\langle x \rangle}}} \frac{1 + 2\tilde{k} - \frac{3\lambda}{\gamma}\tilde{k}^2 + \left(\frac{\lambda}{\gamma}\right)^2 \tilde{k}^3}{\left(1 + \tilde{k} - \frac{\lambda}{\gamma}\tilde{k}^2\right)^{3/2}}, \tag{11c}$$

where $\tilde{k} = k\beta/(1 + k\beta)$ is an auxiliary constant, and these are compared to their counterparts in the single-cell perspectives in Fig 2C. To obtain the limit of weak feedback strength ($k \ll 1$), we take the approach used in (8) and fix $\overline{\langle x \rangle}_{SC}$. The statistics for the protein from population perspective (11) can be expressed in terms of their single-cell counterparts and the parameter $k$:

$$\overline{\langle x \rangle}_{Pop} \approx \overline{\langle x \rangle}_{SC}(1 - k\beta), \tag{12a}$$

$$\left(\mathrm{CV}_x^2\right)_{Pop} \approx \left(\mathrm{CV}_x^2\right)_{SC}(1 + k\beta), \tag{12b}$$

$$(\mathrm{Skew}_x)_{Pop} \approx (\mathrm{Skew}_x)_{SC}\left(1 + k\frac{\beta}{2}\right). \tag{12c}$$

We see that from the population perspective, noise and skewness (given by (12b) and (12c), respectively) increase at least linearly faster than the moments for a single cell. This is because as the feedback intensifies, the population includes more fast-proliferating cells with low protein concentration, and the mean protein level $\overline{\langle x \rangle}_{Pop}$ decreases to zero (Fig 2C, upper panel). This causes higher noise levels for stronger feedback (Fig 2C, middle panel). Finally, as $k$ increases, the population distribution becomes more right-skewed (Fig 2C, lower panel) as a result of population composition shifting to proliferative cells with lower concentrations.

**Strong feedback limit and the dynamics of divergence.** Besides the approximations of weak feedback, we can also study the strong feedback limit, while keeping $\overline{\langle x \rangle}_{SC}$ fixed according to (7). In this limit, the statistics in each perspective show different properties. Noise and skewness in the single-cell framework exhibit the following limits:

$$\lim_{k \to \infty} \left(\mathrm{CV}_x^2\right)_{SC} = 1 + 2\beta/\overline{\langle x \rangle}_{SC}, \tag{13a}$$

$$\lim_{k \to \infty} (\mathrm{Skew}_x)_{SC} = 2\frac{1 - \tilde{\beta}^3}{\left(1 - \tilde{\beta}^2\right)^{3/2}}, \quad \tilde{\beta} = \frac{\beta}{\langle x \rangle + \beta}. \tag{13b}$$

In contrast, these statistics are unbounded in the population perspective, i.e.,

$$\lim_{k \to \infty} \left( \mathrm{CV}_x^2 \right)_{Pop} = \lim_{k \to \infty} \left( \mathrm{Skew}_x \right)_{Pop} = \infty. \tag{14}$$

In a single cell, the existence conditions provide an exact counterbalancing relationship between the feedback strength and the intrinsic noise (stochastic production); thus, the noise is finite. In the population perspective, as we keep $\overline{\langle x \rangle}_{SC}$ fixed, when the feedback increases, only cells with low protein levels proliferate, shifting the mean to zero, making the other moments diverge.

Another context worth studying involves the scenario where the protein distribution can exist in the population perspective but not at the single-cell level (details in S1 Text Sect 3). Using simulations, we explore how the protein levels diverge within this parameter region (Fig A in S1 Text). In this context, the mean protein concentration diverges over time at the single-cell level because dilution cannot counterbalance protein production. However, in the population, this divergence is balanced by the proliferation of cells with lower protein concentrations, maintaining a steady-state distribution at the population level.

**Complementary formulations for protein statistics.**   Biologically, the coupling between gene product levels and proliferation rate can be more complex than the simple model explained here. To have a broader view of this phenomenon, we explored the following scenarios that can provide better biological accuracy in specific contexts: increasing the feedback strength without fixing the mean protein level; assuming that the transcription rate is proportional to the proliferation rate; considering a short-lived protein, and including a cell cycle duration with different levels of randomness.

First, we explore how the protein statistics change when we increase the feedback strength $k$ while keeping the parameters $\lambda$, $\beta$, and $\gamma$ fixed. This means that $\overline{\langle x \rangle}_{SC}$ is not fixed. Under these conditions, in addition to an eventual divergence of $\overline{\langle x \rangle}_{SC}$, we obtained similar results: the noise level and skewness in the single-cell perspective are always lower than in the population one; further details and comparisons with the unregulated case are provided in S1 Text, Sect 4.

As a second scenario, we studied how single cell and population statistics differ if the transcription rate is not constant but depends on the cell growth rate ($\lambda(x) \propto \gamma(x)$) [94]. In this context, we observe that the feedback strength induces smaller differences in the protein means between the single-cell and the population perspectives. The mean at population level is around 10% less than the mean for single cell while the noise is essentially the same between both perspectives. Details of this model are provided in S1 Text Sect 5.

Third, we analyzed a model in which the protein degrades at a constant rate with a time scale shorter than that of the cell cycle duration. The results indicate that as the natural degradation rate surpasses the dilution rate, the protein distributions at both the single-cell and population levels converge to a gamma distribution corresponding to unregulated gene expression. Additional details are provided in S1 Text Sect 6. As a final scenario, we studied the effect of modeling the cell cycle duration to have a distribution tighter than exponential, as observed in most experiments. Using simulations, we observe that the moments of proteins in both perspectives show negligible changes when the duration of the cell cycle is less noisy (Fig E in S1 Text). Additional details are provided in S1 Text Sect 7.

Now, having explored the impact of growth-mediated feedback on protein statistics, we study how additional noise generated during the partitioning of protein molecules between daughter cells impacts protein stochasticity in single-cell and population perspectives.

## Effects of molecule partitioning on protein concentration fluctuations

During cell division or mitosis, a parent cell segregates its contents, including chromosomes, organelles, and gene products, between daughters. In the absence of any active regulation of segregation, each molecule has a random chance of being inherited by each descendant cell. This randomness in partitioning constitutes an additional noise source that drives intercellular heterogeneity in protein levels [67,95–99]. We investigated whether this source of noise predicts differences on expression variability in single-cell and population perspectives. To isolate the effect of partitioning noise, we first explore an unregulated model in which gene expression evolves deterministically rather through stochastic bursting. Although partitioning effects result from the discreteness of molecules, we model protein levels using a continuous variable and consider partitioning as a jump process. This approach allows us to separately add specific sources of noise while neglecting intrinsic fluctuations typically found in discrete models (such as discrete production and degradation of molecules). Rather, any intrinsic source of noise is effectively captured by effective transcriptional bursting. First, we present the solution focusing on partitioning noise, which follows binomial statistics (mean and variance) equivalent to the discrete counterpart [100]. Next, we add intrinsic noise in the form of stochastic bursting events.

**Coupling deterministic expression with partitioning errors.**   As a starting point, we ignore noise in gene expression and the protein concentration $x$ evolves deterministically as per the first-order differential equation:

$$\frac{\mathrm{d}x}{\mathrm{d}t} = \lambda\beta - \gamma x. \tag{15}$$

The term $\lambda\beta$ (the product of the burst frequency and the average burst size) represents the net average protein synthesis rate and $\gamma$ is the constant dilution rate ignoring feedback regulation.

In the previous section, we assume that the cell cycle duration, the time between consecutive divisions, is exponentially distributed. Relaxing this assumption, we now consider the cell-cycle time to be an independent and identically distributed random variable $\tau_d$ that can follow any arbitrary positively-valued continuous distribution with mean $\langle\tau_d\rangle$ and noise quantified through its squared coefficient of variation, $\mathrm{CV}^2_{\tau_d}$ (Fig 3A). We set the mean duration of the cell cycle such that the population doubles every $\ln(2)/\gamma$ units of time. The particular value of $\langle\tau_d\rangle$ satisfying this requirement will depend on the value of $\mathrm{CV}^2_{\tau_d}$ as previously explained [54].

Having defined the timing of cell-division events, we next describe how we model the random partitioning during the process. First, consider a mother cell with concentration $x$ just before division. A division event results in two daughters with concentrations $x^+$ and $2x - x^+$ respectively. Here $x^+$ is a random variable that is appropriately bounded $0 < x^+ < 2x$ to ensure non-negative concentrations (see S1 Text Sect 8), and has the following mean and variance (conditioned on $x$)

$$\langle x_+\rangle = x, \quad \mathrm{Var}(x_+) = \varepsilon x. \tag{16}$$

Note that the mean protein concentration in both newborn daughters is the same as the mother cell. The constant $\varepsilon$ quantifies the extent of randomness in the partitioning process and depends on multiple of factors, such as the cell size at mitosis, the specifics of molecular segregation (for example, molecules segregating as monomers or dimers), errors in cell size partitioning, etc. Note that the scenario of perfect partitioning ($x_+ = x$ with probability one) is

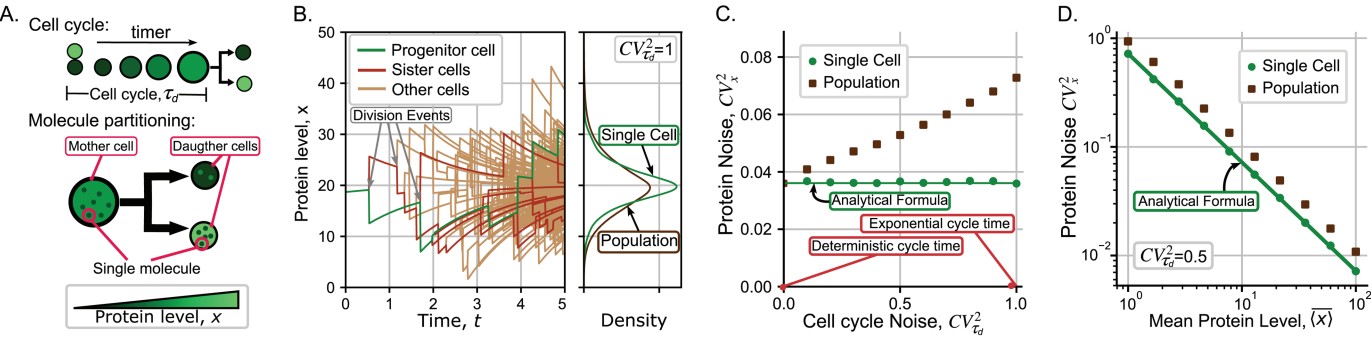

**Fig 3. Noisy cell-cycle durations amplify protein noise differences between single-cell and population perspectives.** (**A**) *(top:)* The cell-cycle duration $\tau_d$ is a random variable following an arbitrary distribution. Within the cell cycle, the protein concentration evolves deterministically as per the ordinary differential equation (15). *(bottom:)* During mitosis, protein molecules are randomly segregated among daughters resulting in differences in the inherited concentration. Different shades of green represent different levels of protein concentration. (**B**) *(left:)* Trajectories of protein concentration in an expanding cell colony, where jumps represent randomness in protein partitioning among daughters during cell division. The green line: a single-cell trajectory is generated by randomly choosing one of the two daughter cells (red lines) after each division event. The light brown lines represent other descendant cells. The cell-cycle times are assumed to be exponentially distributed in this simulation. *(right:)* The steady-state probability density functions of the protein concentration in single-cell and population perspectives. Single-cell statistics are estimated over a 5000 independent individuals; population statistics are estimated using all cells of 2000 colonies (including sisters, progenitor and other cells). Statistics were calculated after 6 generations. (**C**) Effect of noise in the cell cycle time as quantified by its squared coefficient of variation ($CV^2_{\tau_d}$) on the noise in the protein concentration ($CV^2_x$). The solid line is the analytically predicted noise in the single-cell perspective as given by (18), and the dots represent noise levels computed from simulations. Mean concentrations in both models are identical ($\overline{\langle x \rangle}_{SC} = \overline{\langle x \rangle}_{Pop} = \overline{\langle x \rangle} = 20$). (**D**) A logarithmic scale representation of the steady-state protein noise level as a function of the mean protein level, highlighting variability differences between single-cell and population perspectives. Parameters used for the plot are $\gamma = \ln 2$, $\langle \tau_d \rangle = 1$, $\varepsilon = 1$, $k\beta = 20\gamma$.

recovered for $\varepsilon = 0$. We refer the reader to S1 Text Sect 8 for further details on this approach and how $x^+$ is randomly generated.

**Partitioning noise for single-cell perspective.** In the single-cell perspective, $x(t)$ is a PDMP with deterministic dynamics (15) and resets

$$x \mapsto x_+, \tag{17}$$

that occur during the division events (Fig 3B). If the protein level after the division $x^+$ follows the statistics (16), it is possible to obtain exact analytical formulas for the steady-state mean and noise levels of $x(t)$ (see S1 Text Sect 9 for details). More specifically, our results show that for any arbitrarily distributed cell-cycle time $\tau_d$, the protein statistics are

$$\overline{\langle x \rangle}_{SC} = \overline{\langle x \rangle} = \frac{\lambda\beta}{\gamma}, \quad (CV^2_x)_{SC} = \frac{1}{\overline{\langle x \rangle}}\left(\frac{\varepsilon}{2\ln 2}\right), \tag{18}$$

and remarkably, they are invariant of the statistical properties of the cell-cycle time. Thus, making the cell-cycle times more random, this is, increasing $CV^2_{\tau_d}$ for fixed mean $\langle \tau_d \rangle$, will not have any impact on the protein noise level (Fig 3C). Notice the inverse scaling of noise $(CV^2_x)_{SC}$ with the mean protein level in (18) (Fig 3D) is a direct consequence of the variance of $x_+$ being proportional to the concentration in (16). This is also seen in the unregulated bursting model (4) leading to indistinguishability of noise mechanisms from such scaling relationships [67].

**Partitioning noise for population perspective.** Having analytically solved the statistics of concentration fluctuations in the single-cell perspective, we turn our attention to quantifying

protein variability in an expanding cell colony. In the limit case when the cell-cycle duration is fixed (this is $\tau_d = \langle \tau_d \rangle$ with probability one) we show analytically that both perspectives show the same concentration noise (S1 Text Sect 10).

For general randomly distributed cell-cycle duration, we resort to simulation of agent-based models as done in previous works [101,102]. The basis of the simulation algorithm with more details is provided in S1 Text Sect 11 and its implementation is published in our repository [103]. In simulation, we consider each cell as an agent with particular properties, such as protein level $x$ and an internal timer that regulates its division timing. Each division event leads to two newborn cells with concentration partitioning as described above. During the cell cycle, protein concentrations evolve by (15) considering deterministic expression, and the time to the next division is drawn independently according to a prescribed arbitrary statistical distribution.

Sample realizations of protein concentrations from this agent-based framework are illustrated in Fig 3B. Both perspectives yield the same mean concentration while the protein level noise follows:

$$\left( CV_x^2 \right)_{Pop} > \left( CV_x^2 \right)_{SC}. \tag{19}$$

This means that protein has a higher noise in protein concentration from the population perspective. A quantification of this difference is presented in Fig 3C, where both noise levels start out equal when $CV_{\tau_d}^2 = 0$. In contrast to the single-cell perspective (where $\left( CV_x^2 \right)_{SC}$ is invariant of $CV_{\tau_d}^2$), $\left( CV_x^2 \right)_{Pop}$ increases monotonically with increasing randomness in cell-cycle duration. For exponentially-distributed cell-cycle times ($CV_{\tau_d}^2 = 1$), the approximation used by several models [104], and as assumed in the PBE model of the previous section, $\left( CV_x^2 \right)_{Pop}$ is approximately twice of $\left( CV_x^2 \right)_{SC}$ (Fig 3C). The inverse scaling of noise with mean as seen in (18) is also preserved in the population perspective, although with a higher proportionality constant resulting in a shifted line in the logarithmic scale (Fig 3D) (similar to that seen in (12b) for the case of dilution-based feedback). Observe that if there is no partitioning noise, the cell cycle noise will not affect the protein noise. This occurs since the main drivers of heterogeneity in protein levels (cycle-dependent injections of noise) are not present for that case.

Finally, the qualitative differences seen in Fig 3C are recapitulated in more realistic agent-based models that explicitly take into account cell size dynamics (see Fig G in S1 Text and S1 Text Sect 12), and here cell size homeostasis mechanisms drive mother-daughter and daughter-daughter cell-cycle correlations.

**Coupling stochastic expression with partitioning errors.** To complete the approach, we now consider stochastic gene expression as captured by protein synthesis occurring in random bursts. From a single-cell perspective, the protein noise is analytically derived as (see details in S1 Text Sect 9)

$$(CV_x^2)_{SC} = \frac{1}{\langle x \rangle} \left( \frac{\varepsilon}{2 \ln 2} + \beta \right), \tag{20}$$

and is the sum of two noise contributions as given by equations (4) (contribution form the transcriptional bursting noise) and (18) (contribution form partitioning noise). To obtain the corresponding statistics in the population perspective we modify the earlier described agent-based model to consider intracellular concentrations evolving via stochastic bursts and continuous exponential decay with a constant dilution rate $\gamma$ in between bursts. Fig 4 shows simulation trajectories corresponding to two different scenarios:

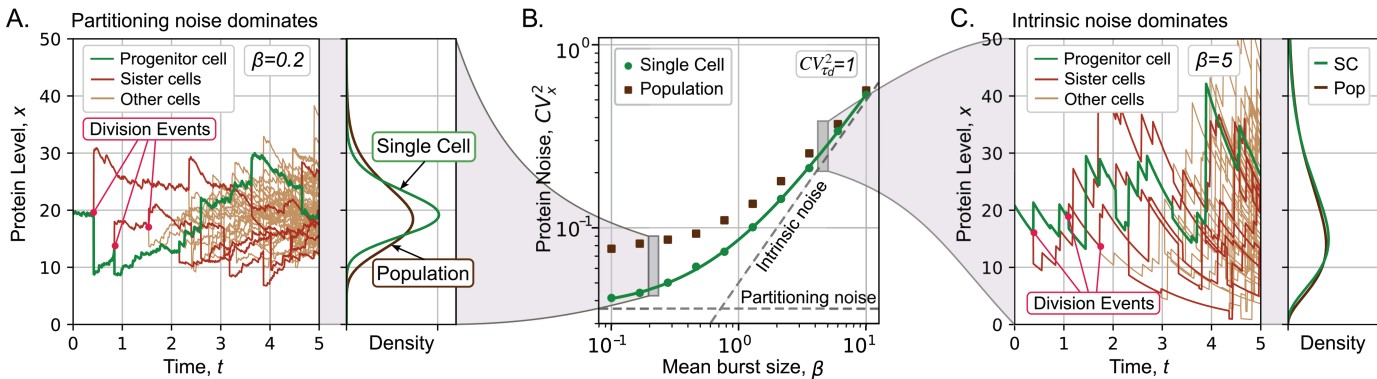

**Fig 4. Increasing randomness in molecular segregation between daughters enhances protein noise differences between single-cell and population perspectives**. (A) Sample trajectories of protein concentration in an expanding cell colony when expression variability is dominated by partitioning noise ($\beta = 0.2$ and $\varepsilon = 1$ in (20)). (B) Comparison of the steady-state protein concentration noise ($CV_x^2$) from single-cell (green circles) and population perspectives (brown squares) calculated from simulations of the agent-based model. The solid line represents the analytically-predicted noise level (20). (C) Sample concentration trajectories for the high intrinsic noise scenario ($\beta = 5$ and $\varepsilon = 1$ in (20)). Other parameters are taken as $\gamma = \ln 2$, $\langle x \rangle = 20$, $\lambda = \gamma \langle x \rangle / \beta$, $\varepsilon = 1$). 2000 colonies where simulated for population perspective, 5000 individuals where simulated for single-cell perspective.

- One where the partitioning noise dominates ($\beta \ll \varepsilon$), in which case $\left( CV_x^2 \right)_{Pop} > \left( CV_x^2 \right)_{SC}$ (Fig 4A).
- The other where intrinsic noise dominates ($\beta \gg \varepsilon$), in which case $\left( CV_x^2 \right)_{Pop} \approx \left( CV_x^2 \right)_{SC}$ (Fig 4C).

Fig 4B quantifies these differences with increasing intrinsic noise component, i.e., increasing $\beta$. The key message of this figure is that when intrinsic noise dominates, then both perspectives are similar in terms of the concentration statistics. This can be intuitively understood from our earlier analytical results where both perspectives yield similar protein concentration pdfs (Fig 2B) in the case of perfect partitioning ($\varepsilon = 0$). In contrast, the gap between $\left( CV_x^2 \right)_{Pop}$ and $\left( CV_x^2 \right)_{SC}$ increases as partitioning noise begins to dominate.

## Discussion

In this manuscript, we have investigated stochastic concentration fluctuations in an individual cell over time (single-cell perspective) and across all descendant cells at a fixed time point (population perspective). A key assumption is that the protein of interest is long-lived; thus, its decay is dominated by dilution from cellular growth. We identified two scenarios where the concentration statistics are different between single-cell and population perspectives:

- Expression-growth coupling, where a cell's proliferation rate depends on the concentration of a specific protein.
- Random partitioning of molecules between daughters during cell division.

Consistent with previous observations [53], our analytical results corroborated with the simulation of agent-based models find that the noise in the single-cell perspective is underestimated (Figs 2C and 3C). We discuss these scenarios in more detail below. The reduction in cellular growth rate with increasing protein concentration is captured phenomenologically through expression (2). This expression defines the feedback in gene expression, where bursts in protein synthesis represent the intrinsic noise in gene expression. This feedback has

been observed in specific cases, including stress-response factors in bacteria such as RpoS [58] and others in fission yeast [105]. These factors regulate the transcription of other genes, thus establishing an indirect relationship between gene products and the growth rate. Our approach can also be applied to more direct examples, such as proteins with significant toxicity levels that inhibit cell growth [61]. This scenario is particularly relevant in the case of synthetic constructs where the gene product is non-functional to the cell. Other proteins that affect cell proliferation, such as division timing regulators, may require a more complex cell cycle-dependent modeling [106].

The continuous protein dilution between consecutive bursts is defined by the differential equation (3). We derive exact analytical formulas for steady-state concentration probability density functions at both single-cell and population perspectives, as given by (5) and (10), respectively. Before discussing these results, we comment on the no-feedback case. In the absence of any concentration-dilution coupling, both perspectives yield the same gamma-distributed concentration. This result can be generalized to consider transcription feedbacks that are common regulatory features in gene expression [107,108]. Our analysis (see S1 Text Sect 13) shows that a constant dilution rate with an arbitrary concentration-dependent burst frequency yields identical concentration distributions in both perspectives.

The presence of dilution-based positive feedback shifts the concentration distribution in the population perspective to lower protein levels (Fig 2B) due to increased proliferation of low-expressing cells. This explains the lower mean levels and the higher noise and skewness in cell populations compared to the single-cell perspective (Fig 2C). A key qualitative difference is seen in the parameter space where the steady-state concentration distributions exist (Fig 2A). In the single-cell perspective this existence region is defined by the net synthesis rate ($\lambda\beta$) being lower than the maximum rate of concentration decrease $\gamma/k$ in (3) that is reached when the protein level is high. The existence region is expanded in the population perspective as cells with higher concentrations proliferate slower, and hence do not contribute significantly to the population.

Although we highlighted the differences between single-cell and population approaches, it is also important to mention the scenarios where single-cell statistics are similar to population statistics. First, we found:

- When the protein half-life is short relative to the cell cycle (Fig D in S1 Text).

In the context of protein half-life being longer relative to the cell cycle, we observe additional regimes:

- When partitioning noise is much smaller than intrinsic noise and the protein level does not affect cell growth or proliferation.
- When the burst frequency is proportional to the growth/dilution rate. Within the explored parameter regime, the differences between both approaches are relatively small (Fig C in S1 Text).

The equivalence of these scenarios is relevant for experimental design. In these scenarios, it is possible to infer the protein statistics of one perspective measuring the other. Our work offers a framework for estimating the degree of similarity between these population perspectives.

The approach of the article can be expanded in multiple ways. While we have kept the modeling framework simple to obtain analytical insights, our models can be refined in the future to consider the scaling of expression rates with the dilution rate [94], explicitly

accounting for cell size and cell-cycle effects [109–111], incorporating promoter transcriptional states and mRNA dynamics in more complex gene expression models [112]. Other modifications of the model can include a negative feedback in the proliferation rate. In a previous contribution, for instance, we investigated dilution-based negative feedback [102], where increasing concentration increases the cellular growth (dilution) rate. This would be the case for many cellular growth factors, where lower concentrations result in lower proliferative capacities [113]. As expected, the results here are opposite to those seen in Fig 2, with the distribution now shifting to a higher concentration in the cell population relative to the single-cell distribution [102]. *In summary, if the expression of a specific gene determines cellular proliferative capacity at an individual-cell level, then the statistical fluctuations in its gene product levels can be qualitatively and quantitatively different between population and single-cell perspectives.*

The discrepancy between single-cell and population perspectives also arises when considering another source of intrinsic noise, the random segregation of molecules between daughters. In the single-cell framework, these random segregation events appear as *jumps* or noise-injections in the concentration at division times (Fig 3B). The statistical properties of these jumps are defined in (16) where the degree of partitioning noise can be tuned through the variable $\varepsilon$. We derived the steady-state noise in concentration (20) for an arbitrarily distributed cell-cycle duration and find it to be insensitive to fluctuations in cell-cycle times. In contrast, the corresponding noise within a cell population increases monotonically with increasing randomness in cell-cycle times (Fig 3C). One way to explain this effect is that the randomness in cell cycle times manifests itself as a fluctuation in colony size [114], and larger colonies exhibit higher intracellular concentration fluctuations resulting from the accumulation of noise-injecting division events. Thus, while both perspectives predict similar noise levels for a fixed cell cycle duration, the noise gap increases with $\mathrm{CV}^2_{\tau_d}$, and the noise at the population level is approximately twice the noise in single cells when the cell cycle timing is distributed exponentially (Fig 3C).

A key limitation of our modeling approach is that cell-cycle duration is considered to be *timer*, that is, each duration is independently and identically distributed. It is well known that, if cells grow exponentially in cell size along the cell cycle, then such timer-based models are not able to provide cell size homeostasis, that is, the variance in cell size grows unboundedly over time [115,116]. We address this limitation by modifying the agent-based model to explicitly consider the size dynamics of individual cells and implemented size control according to *adder* – the size added from cell birth to division is not correlated with the newborn size [117–122]. Statistics computed from simulating these cell-size homeostatic models are presented in S1 Text Sect 12 and recapitulate the qualitative finding of Fig 3C: protein noise in a cell population is more sensitive to fluctuations in added size compared to the single-cell perspective. Interestingly, these simulations show that the single-cell concentration noise that is invariant to $\mathrm{CV}^2_{\tau_d}$ in the timer model (Fig 3C), increases slightly with $\mathrm{CV}^2_{\tau_d}$ in the adder model (Fig G in S1 Text). The agent-based models used in this study have been uploaded to *zenodo* for the research community and can be modified to include other types of size control mechanisms and more complex biochemical processes of gene expression [103].

In summary, we have determined differences in studying gene expression in isolated cells versus expanding cell lineages. Although we specifically considered stable proteins, these results can be adapted to other types of biomolecules, such as mRNAs & metabolites, and extended to study intracellular differences in chromosome abundance, plasmids and organelles [123–125]. The focus on intrinsic noise mechanisms (bursting and partitioning noise) can also be generalized to explore extrinsic noise through parametric fluctuations in

gene product synthesis/decay rates [126,128,129]. For example, recent work has reported random fluctuations in translation rates in *Schizosaccharomyces pombe* that dissipate quickly within a cell cycle [130]. Finally, it will be interesting to test our predictions with single-cell expression data using lineage tracking via cellular barcodes, where the extent of cell proliferation can be directly linked to gene expression patterns and their corresponding statistical fluctuations [64].

## Methods

### Analytical methods

The single-cell model is formulated as a piecewise-deterministic Markov process with deterministic growth-mediated dilution and stochastic jumps (bursts). In general form, the deterministic component is represented by a dynamical system $\dot{x}(t) = A(x(t), t)$. Stochastic bursts are characterized by a burst kernel $\omega(b|y, t)$, which describes the probability of transitioning from state $y$ to $x = y + b$ within an infinitesimal time interval. Bursts occur at a rate $\lambda(x, t)$, with their sizes drawn from an arbitrary non-negative distribution.

The time evolution of the protein probability density at the single-cell level, $p_{SC}(x, t)$, is described by the Chapman-Kolmogorov equation:

$$\frac{\partial}{\partial t} p_{SC}(x, t) = -\frac{\partial}{\partial x}\left(A(x, t)p_{SC}(x, t)\right) - \lambda(x, t)p_{SC}(x, t)$$
$$+ \int \lambda(y, t)\omega(x - y|y, t)p_{SC}(y, t)\mathrm{d}y.$$

In this work, the burst frequency is constant ($\lambda(x, t) \equiv \lambda$) and bursts are exponentially distributed ($\omega(b|y, t) = e^{-(x-y)/\beta}/\beta$) as per (1), while the dynamical system $A(x,t)$ is given by (3). Under these assumptions, the Chapman-Kolmogorov equation becomes:

$$\frac{\partial p_{SC}(x, t)}{\partial t} = \frac{\partial}{\partial x}\left(\frac{\gamma x}{1 + kx}p_{SC}(x, t)\right) + \frac{\lambda}{\beta}\int_0^x e^{-(x-y)/\beta}p_{SC}(y, t)\mathrm{d}y - \lambda p_{SC}(x, t), \qquad (21)$$

the solution of which is provided in S1 Text Sect 1.

To extend the model to the population level, we describe the cell cycle mechanism, where at the end of the cycle, a mother cell is replaced by two daughter cells. The daughters inherit the mother's protein concentration and half of her cell volume, leading to a sequence of Markov processes; the model for each individual cell remains as described previously. We also introduce the net rate of particle generation (population growth), $G(x,t)$, which captures the combined effects of all birth-death processes. Here, we further assume that the net growth rate is proportional to the number of particles in state $x$, i.e. $G(x, t) = \gamma(x, t)h(x, t)$.

The time evolution of the population is governed by the population balance equation:

$$\frac{\partial}{\partial t}h(x, t) = \frac{\partial}{\partial x}\left(\frac{\gamma x}{1 + kx}h(x, t)\right) + \frac{\gamma}{1 + kx}h(x, t)$$
$$+ \frac{\lambda}{\beta}\int_0^x e^{-(x-y)/\beta}h(y, t)\mathrm{d}y - \lambda h(x, t), \qquad (22)$$

where $h(x,t)$ is the expected population density and describes the number of cells with concentration $x$ at time $t$. We assume that $h(x,t)$ can be separated as $h(x, t) = e^{\mu t}p_{Pop}(x)$, where $\mu$ is the population growth rate and $p_{Pop}(x)$ is the protein distribution at the population level. The solution of (22) is provided in S1 Text Sect 2.

### Simulation methods

To estimate protein statistics considering partitioning, we use an agent-based algorithm, modeling protein levels within each cell over time. For the single-cell perspective, only one descendant is chosen during each division, while in the population model, a new cell is added after each division, allowing the population to grow. We used two methods of simulation. Simulations for the case of feedback in dilution were performed using the tau-leaping algorithm [131]. For the section of partitioning noise, we used a modification of the Gillespie method [132]. Each cell tracks the time to the next burst and division, with burst times exponentially distributed and division times gamma distributed. During each iteration, the minimum time until the next reaction is selected, and the respective reaction occurs. Protein levels evolve according to a differential equation, with distinct behaviors for bursty and non-bursty protein synthesis. Additional details of the algorithm are shown in the S1 Text, Sect 11.

## Artificial intelligence

The authors declare that they did not use generative AI or AI-assisted technologies in the writing process.

## Supporting information

**S1 Text. Supporting information.**
(PDF)

## Author contributions

**Conceptualization:** Cesar Nieto, Pavol Bokes, Abhyudai Singh.

**Data curation:** Zhanhao Zhang, Iryna Zabaikina, Cesar Nieto.

**Formal analysis:** Zhanhao Zhang, Iryna Zabaikina, Zahra Vahdat.

**Funding acquisition:** Pavol Bokes, Abhyudai Singh.

**Investigation:** Zhanhao Zhang, Iryna Zabaikina, Cesar Nieto.

**Methodology:** Zhanhao Zhang, Iryna Zabaikina, Zahra Vahdat.

**Project administration:** Pavol Bokes, Abhyudai Singh.

**Resources:** Pavol Bokes, Abhyudai Singh.

**Software:** Zhanhao Zhang, Iryna Zabaikina, Cesar Nieto.

**Supervision:** Pavol Bokes, Abhyudai Singh.

**Validation:** Cesar Nieto, Pavol Bokes, Abhyudai Singh.

**Visualization:** Zhanhao Zhang, Iryna Zabaikina, Cesar Nieto.

**Writing – original draft:** Zhanhao Zhang, Iryna Zabaikina, Cesar Nieto, Zahra Vahdat.

**Writing – review & editing:** Cesar Nieto, Pavol Bokes, Abhyudai Singh.

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
