## [Decision Letter · Decision Letter 0]

16 Sep 2024

Dear Dr Nieto,

Thank you very much for submitting your manuscript "Stochastic Gene Expression in Proliferating Cells: Differing Noise Intensity in Single-Cell and Population Perspectives." for consideration at PLOS Computational Biology.

As with all papers reviewed by the journal, your manuscript was reviewed by members of the editorial board and by several independent reviewers. In light of the reviews (below this email), we would like to invite the resubmission of a significantly-revised version that takes into account the reviewers' comments.

We cannot make any decision about publication until we have seen the revised manuscript and your response to the reviewers' comments. Your revised manuscript is also likely to be sent to reviewers for further evaluation.

Sincerely,

Oleg A Igoshin

Academic Editor

PLOS Computational Biology

Marc Birtwistle

Section Editor

PLOS Computational Biology

Reviewer's Responses to Questions

**Comments to the Authors:**

Reviewer #1: Review uploaded as an attachment.

Reviewer #2: This is an interesting paper examining the differences between noise in protein concentration when measured across lineages, and across the population. It has been shown previously (eg in the work of Thomas, et al cited in the paper) that these two ways of determining variability can lead to different results. This manuscript extends these previous observations by presenting analytical results in the case that protein concentration inhibits growth.

The results are clearly presented, and should be of interest to people working in this field. I recommend that it be published. However, addressing the following comments would improve the manuscript:

I was not able to understand a point that is fundamental to the claims in the paper: In the single cell model divisions are not represented explicitly. If I understand correctly this is because the concentration is continuous across divisions, and thus divisions have not impact on it. On the other hand, in the population model divisions need to be represented, and they do seem to play an essential role. Therefore, I do not think that we can view the single cell model as a single lineage in the population model. This is in contrast to the work of Philipp Thomas, for example, which does contrast the statistics across a lineage and across a population.

This also population distribution exists even when the lineage distribution does not. I do not see how it would be possible that across all lineages the protein levels diverge, in such a way that the population distribution remains stable. I assume something like this is possible, but - if this is what is really claimed - should be explained. In short, the relation between the two models should be clarified a bit more, and the relation to the work of lineage vs population statistics (such ss that of Thomas), should be clarified.

Some minor comments:

In the intro and discussion, it is a little unclear that the authors are talking about lineages. Unless reading carefully, it may see like the discussion is about a single division cycle.

I also have some questions about the notation: Why are angular brackets used along with a bart to denote the average (eg line 124 and equations below).

A brief discussion of what happens in the limit where only one protein is produced at a time would be interesting (ie what would be missed if bursts were ignored).

Figure 1a, top: What do the arrows pointing up mean?

The manuscript could use some editing for language. There are no glaring grammatical errors, but there are many odd phrasings. Improving the presentation would make for an easier reading experience. For example:

- single-cell studies have interestingly revealed

- the second scenario is when

- Line 319: “corresponding”, not “corrupting.”

Reviewer #3: I commend the authors for submitting a proofread article with carefully made figures.

Review is uploaded as an attachment.

**Have the authors made all data and (if applicable) computational code underlying the findings in their manuscript fully available?**

Reviewer #1: Yes

Reviewer #2: **No: **I didn't see the code, but may have missed it. But the simulations are

fairly straightforward.

Reviewer #3: **No: **The code to run the simulations is provided, but the simulated data is not, nor is the code to analyse the simulated data.

PLOS authors have the option to publish the peer review history of their article (what does this mean?). If published, this will include your full peer review and any attached files.

Reviewer #1: No

Reviewer #2: No

Reviewer #3: **Yes: **Nataša Puzović
---

## [Decision Letter · Decision Letter 1]

19 Feb 2025

PCOMPBIOL-D-24-01119R1

Stochastic Gene Expression in Proliferating Cells: Differing Noise Intensity in Single-Cell and Population Perspectives.

PLOS Computational Biology

Dear Dr. Nieto,

Thank you for submitting your manuscript to PLOS Computational Biology. After careful consideration, we feel that it has merit but does not fully meet PLOS Computational Biology's publication criteria as it currently stands. Therefore, we invite you to submit a revised version of the manuscript that addresses the points raised during the review process.

Please submit your revised manuscript within 30 days Apr 21 2025 11:59PM. If you will need more time than this to complete your revisions, please reply to this message or contact the journal office at ploscompbiol@plos.org. Please include the following items when submitting your revised manuscript:

We look forward to receiving your revised manuscript.

Kind regards,

Oleg A Igoshin

Academic Editor

PLOS Computational Biology

Marc Birtwistle

Section Editor

PLOS Computational Biology

**Additional Editor Comments:**

Please address remainign minor comments of the Reviewer 1 and we should be able to accept the paper after that/

**Journal Requirements:**

At this stage, the following Authors/Authors require contributions: Cesar Nieto, and Abhyudai Singh. Please ensure that the full contributions of each author are acknowledged in the "Add/Edit/Remove Authors" section of our submission form.

2) Please amend your detailed Financial Disclosure statement. This is published with the article. It must therefore be completed in full sentences and contain the exact wording you wish to be published.

3) Please ensure that the funders and grant numbers match between the Financial Disclosure field and the Funding Information tab in your submission form. Note that the funders must be provided in the same order in both places as well. Currently, the order of the grants is different in both places.

Please indicate by return email the full and correct funding information for your study and confirm the order in which funding contributions should appear. Please be sure to indicate whether the funders played any role in the study design, data collection and analysis, decision to publish, or preparation of the manuscript.

**Reviewers' comments:**

Reviewer's Responses to Questions

**Comments to the Authors:**

**Please note that one of the reviews is uploaded as an attachment.**

Reviewer #1: Review uploaded as an attachment

Reviewer #2: The authors have addressed my comments, and those of the other referees.

Reviewer #3: The authors have addressed all of my questions and made adequate adjustment to the manuscript.

**Have the authors made all data and (if applicable) computational code underlying the findings in their manuscript fully available?**

Reviewer #1: Yes

Reviewer #2: Yes

Reviewer #3: Yes

PLOS authors have the option to publish the peer review history of their article (what does this mean?). If published, this will include your full peer review and any attached files.

Reviewer #1: No

Reviewer #2: No

Reviewer #3: **Yes: **Nataša Puzović

**Figure resubmission:**
---

## [Editor Report · Decision Letter 2]

31 Mar 2025

Dear Dr Nieto,

We are pleased to inform you that your manuscript 'Stochastic Gene Expression in Proliferating Cells: Differing Noise Intensity in Single-Cell and Population Perspectives.' has been provisionally accepted for publication in PLOS Computational Biology.

Best regards,

Oleg A Igoshin

Academic Editor

PLOS Computational Biology

Marc Birtwistle

Section Editor

PLOS Computational Biology

---

## [Editor Report · Acceptance letter]

PCOMPBIOL-D-24-01119R2

Stochastic Gene Expression in Proliferating Cells: Differing Noise Intensity in Single-Cell and Population Perspectives.

Dear Dr Nieto,

I am pleased to inform you that your manuscript has been formally accepted for publication in PLOS Computational Biology. Your manuscript is now with our production department and you will be notified of the publication date in due course.

With kind regards,

Anita Estes
